# Exogenous Methyl Jasmonate (MeJA) Improves ‘Ruixue’ Apple Fruit Quality by Regulating Cell Wall Metabolism

**DOI:** 10.3390/foods13111594

**Published:** 2024-05-21

**Authors:** Xiaoyi Ding, Bin Wang, Yubo Gong, Xueqing Yan, Xinxin Chen, Yuanwen Zhong, Zhengyang Zhao

**Affiliations:** State Key Laboratory of Crop Stress Biology for Arid Areas, College of Horticulture, Northwest A&F University, Xianyang 712100, China; dxy091399@163.com (X.D.); wb765198@163.com (B.W.);

**Keywords:** *Malus domestica*, MeJA, *MdPL18*, fruit quality, cell wall metabolism

## Abstract

‘Ruixue’ apples were used as the test material to study the effect of 10 μM methyl jasmonate (MeJA) on the quality and cell wall metabolism of apples after 18 d of storage. The results showed that MeJA significantly decreased the respiratory rate, reduced the titratable acid content and maintained a high soluble solids content. MeJA has been shown to suppress the activities and gene expressions of WSP, CSP, ISP, and cellulose in contrast to the control group, thereby maintaining a lower cell permeability and higher exocarp firmness. MeJA significantly decreased the expression of *MdACS*, *MdACO*, *MdPL*, *Mdgal*, and *MdPG* genes in the apple exocarp when compared to the control group. In addition, the overexpression of *MdPL18* increased the content of cell wall polysaccharides such as WSP and CSP, enhanced cell wall-degrading enzyme activities, and accelerated fruit ripening and softening, whereas silencing *MdPL18* did the opposite. Together, these results demonstrate that exogenous MeJA maintains the Ruixue apple fruit quality by regulating the metabolism of cell wall substances.

## 1. Introduction

Apples belong to the Rosaceae plant family and are the most widely grown common fruit in China [1]. In 2022, China’s apple production and cultivated area accounted for 56.4% and 42.7% of the global total, respectively. An apple is a typical respiratory climacteric fruit, and the fruit quality is best for fresh consumption when it reaches the peak of the respiratory climacteric. Climacteric fruits show a significant increase in ethylene release during the climacteric, and they respond to exogenous ethylene before the climacteric period, during the ripening process [2].

During the processes of harvest and storage, fruits undergo a series of physiological changes, such as changes in their color, texture, and volatile substance content [3]. These changes are influenced by various environmental factors, biological and abiotic stresses, as well as genetic regulation [4]. Ethylene can promote fruit ripening, regulate gender differentiation, and respond to biotic and abiotic stresses [5]. Studies have found that exogenous ethylene treatment can increase cell membrane permeability in papaya fruit and kiwifruit, accelerating fruit ripening and softening [6]. Ethylene treatment can also accelerate the conversion of nutrients in apple fruit, promoting its ripening [7]. From the above, it is clear that exogenous ethylene treatment can improve cell membrane permeability, promote fruit color change, and improve the flavor and conversion of nutrients in fruits.

Extensive research has shown that under the action of cell wall metabolism-related enzymes, the structure of the cell wall, which plays a supportive role, changes, the permeability of the cell is increased, the cellular fluid exudes, and the firmness of the fruit decreases, thus promoting fruit ripening [8]. Throughout the process of fruit softening, alterations occur in the structure of the cell walls, as well as the breakdown of various components within the cell walls [9]. Pectin, cellulose, and hemicellulose are randomly distributed in the cell wall polysaccharide network structure through hydrogen bonds, covalent bonds, and hydrophobic forces [10]. The distribution of pectin in fruits plays a crucial role in determining their texture. As fruits ripen, insoluble protopectin breaks down, while soluble pectin levels rise. This results in reduced intercellular adhesion and structural damage to the cells. These processes are often associated with the involvement of cell wall degradation-related enzymes, such as polygalacturonase (PG), pectin methylesterase (PME), and glycosidase (β-Gal) [11].

Jasmonates (JAs) include jasmonic acid (JA), its volatile ester derivative methyl jasmonate (methy-jasmonate, MeJA), and other amino acid derivatives [12]. They are a class of growth-regulating substances widely present in a variety of plants and can act as endogenous signaling molecules, inducing the expression of plant defense genes, regulating plant resistance, growth and development, and the synthesis of secondary metabolites [13]. They can also affect pollen development and fruit maturation and development [14]. Exogenous MeJA regulates ascorbic acid and glutathione metabolism, thereby improving the quality of loquat fruit [15]. Similarly, the content of ascorbic acid in carambola significantly increases after MeJA treatment. Moreover, it effectively hinders the function of PAL, POD, and PPO, ultimately preserving the ideal fruit condition during transport and preservation. [16].

The firmness index of the fruit is a crucial factor in assessing fruit ripeness. It was found that MeJA treatment could inhibit the decrease in firmness of peach and strawberry fruits [17,18]. Titratable acid is one of the important indicators for evaluating a fruit’s nutritional quality. The treatment of peach and blackberry fruits with MeJA resulted in a decrease in the titratable acids content [17]. Treatment with external MeJA can elevate the levels of volatile aromatic compounds in strawberries, thus enhancing their fragrance and boosting the strawberry quality. [18]. In addition, jasmonic acid also plays an important role in delaying the browning of fresh-cut fruits. MeJA treatment reduces the occurrence of browning in fresh-cut apples.

Pectin lyase is crucial in floral organ development and fruit ripening, serving various biological functions in plants [19,20,21]. Notably, it aids in fruit softening, while pectin-related lyase contributes to polysaccharide modification during ripening [22]. Previous research identified two distinct pectin lyase genes, PelⅠ and PelⅡ, in bananas and showed that their expression levels rise as the fruit matures [23]. In tomatoes, suppressing the pectin lyase gene enhances the fruit’s firmness, delaying softening without compromising the growth or quality [24]. Similarly, in non-climacteric strawberries, pectin lyase gene expression increases with fruit maturation [25]. Silencing this gene in strawberries inhibits endogenous pectin lyase gene expression, leading to pectin accumulation and a 30% increase in the fruit hardness compared to wild-type fruit [25]. These findings suggest pectin lyase as a potential candidate gene for regulating fruit texture.

‘Ruixue’ is a new apple variety with white flesh, a green surface, crispy and sweet taste, and rich flavor. In this research, the effects of exogenous MeJA on fruit quality indicators such as the firmness, SSC, TA, and ETH production of the ‘Ruixue’ apple fruit were studied. Furthermore, the changes in cell wall substances and related enzyme activities of apple fruit after MeJA treatment were further studied. The function of the transcription factor *MdPL18* in the apple fruit was substantiated through transient expression. Overall, MeJA affects the postharvest quality of the ‘Ruixue’ apple through the metabolism of the cell wall.

The objective of this study is to investigate the impact of MeJA on the taste quality, nutritional composition, and cell walls of apples throughout storage after harvesting. Through the examination of the enzyme activities related to cell walls, the effect of treating apple fruit with MeJA was studied. The expression patterns of fruit ripening-related genes were further determined to elucidate the effects of MeJA on the postharvest quality of ‘Ruixue’ apples.

## 2. Materials and Methods

### 2.1. Experimental Treatments and Plant Materials

The variety number of ‘Ruixue’ is CNA20151469.1. During our initial investigation, the ‘Ruixue’ apples underwent treatment with MeJA at concentrations of 10 μM, 100 μM, and 1000 μM, respectively (Appendix A). The results showed that the treatment with 10 μM MeJA had a significant effect on delaying the postharvest ripening of the apples, while the treatment with 100 μM MeJA did not significantly affect the quality of the apples, and the treatment with 1000 μM MeJA accelerated the ripening of the apples. In addition, according to the study by Lv et al. [26], the postharvest treatment of apple fruit with 10 μM MeJA alleviated the postharvest ripening. Hence, for this investigation, we selected a concentration of 10 μM MeJA (Figure 1A). Two groups were formed, with a total of 240 ‘Ruixue’ apples that were uniform and undamaged. The preparation of the MeJA solution involved dissolving 95% MeJA (Solarbio, Beijing, China) in deionized water with a 0.077% (*v*/*v*) triton x-100 solution. The first group was immersed in a solution of 10 μM MeJA for 10 min. The control group, on the other hand, was immersed in deionized water with a 0.077% (*v*/*v*) triton x-100 solution for 10 min. Following air drying, all the apples were stored at a temperature of 20 ± 0.5 °C, and samples were collected at intervals of 3 days (0, 3, 6, 9, 12, 15, and 18). During every sampling occasion, we randomly picked 15 apples from each group. Each treatment had three repeated experiments, with five apples for each replicate. The pericarp tissue (3–8 mm below the equator) of the control and MeJA-treated fruits were collected.

### 2.2. Detection of Physiological Characteristics

During the period of 18 days, the percentage of weight loss for two sets of fruits was assessed every 3 days. The weight reduction was determined by utilizing the subsequent equation:Weight loss (%) = (initial weight − final weight)/initial weight ∗ 100

The collection of ethylene was conducted using the drainage gas collection method. The ethylene release rate was analyzed using gas chromatography (GC-14A, Shimadzu, Kyoto, Japan). The respiration rate was measured using gas chromatography with a flame ionization detector (FID) (SP-9890, Lunan Ruihong, Tengzhou, China). The release of ethylene and the respiration rate were measured in terms of the amount of CO_2_ and ethylene generated per kilogram of fresh weight in an hour, respectively. Each experimental procedure was performed three times, and each replication consisted of 10 pieces of fruit. The hardness of the apple fruit was evaluated by taking measurements at two points equidistant around the circumference of the apples. A texture analyzer (GS-15, Ibbenbüren, Germany) was utilized for the analysis, employing a 10 mm diameter flat probe that could penetrate to a depth of 8 mm. SSC and TA were analyzed as described by yang et al. [27]. The sugar content of the fruit was measured using a digital refractometer. To detect titratable acids, 1.0 g of apple fruit was mixed with 3.0 mL of distilled water, ground, and transferred to a 50 mL volumetric flask. After filtering the solution for 10 min, 5 mL of the filtrate was placed in an Erlenmeyer flask. Then, 2 drops of 2% phenolphthalein reagent were added, followed by titration with NaOH. The calculation was performed using the given formula. In the formula, V is the total volume of the extracted sample liquid, mL; c is the NaOH titrant concentration, mol/L; ΔV is the difference between the volume of NaOH used to titrate the sample and the volume of NaOH used to titrate distilled water, mL; Vs is the volume of the filtrate used in the titration, mL; m is the sample weight, g; and the conversion coefficient *f* = 0.067.
TA(%) = (V × c × ΔV × ƒ)/(Vs × m) × 100

### 2.3. Determination of Cell Wall Component Content

The procedure to extract cell wall polysaccharides followed that of Melton et al. [28]. By utilizing varying solubility levels of cell wall components, the contents of apple fruit cell wall constituents were isolated and analyzed. A sequential extraction was conducted to obtain WSP, CSP, ISP, and hemicellulose. The pectin concentration was assessed using the carbazole–ethanol technique, the hemicellulose concentration was measured using the anthrone colorimetric approach, and the cellulose concentration was determined using the weight-based method.

### 2.4. Determination of Cell Wall Degradation-Related Enzyme Activities

PG activities were determined using the DNS colorimetric method, which was slightly modified from our previous study [29]. The measurement of β-gal activity was performed through the hydrolysis of the nitrogalactose side chain and was quantified at a wavelength of 540 nm. The determination of PL activity was carried out following the protocol outlined by Payasi, with quantification performed at 550 nm [30]. The measurement of β-gal activity was conducted using the method described by Payasi, and quantification was performed at 550 nm [30].

### 2.5. Determination of Relative Gene Expression

The apple fruit samples were processed for total RNA extraction using the Quick RNA isolation kit (Tiangen, Beijing, China). The cDNA reverse transcription was carried out using the All-in-One First-Strand SuperMix with gDNA Eraser (TransGen Biotech, located in Beijing, China). The RT-qPCR analysis was performed with the ABI7500 System and SYBR Green Master Mix (Vazyme from Nanjing, China). All gene expression levels were normalized using *Mdβ-Actin* as the internal reference gene. The relative gene expression was calculated by 2^−ΔΔCt^. The RT-qPCR primers can be found in Appendix A.

### 2.6. Transient Expression of MdPL18 in Apple Fruit

An overexpression vector, pCAMBIA 2300, was utilized to introduce the full-length coding DNA sequence (CDS) of *MdPL18*. As for the silencing vector, a 400 bp fragment of the *MdPL18* CDS was introduced into TRV2 (virus-induced gene silencing). To augment the control, ‘Ruixue’ fruit was injected with A. tumefaciens (EHA105) co-cultured with OE-*MdPL18* and TRV-*MdPL18* using the empty vector (OD_600_ = 1, 100 µM acetosyringone, 100 mM MgCl_2_, and 100 mM MES). Following a 3-day infiltration period, qRT-PCR was performed on samples collected from the injection site. Three biological replicates were carried out, with each treatment involving 20 apples that were subjected to injection.

### 2.7. Stable Overexpression of MdPL18 in Apple Calli

The overexpression vector for *MdPL18* CDS was incorporated into pCAMBIA2300 and utilized. ‘Orin’ apple calli (12 days old, OD_600_ = 0.8) were co-cultured with A. tumefaciens (EHA105) containing OE-*MdPL18*. These calli were then placed in MS medium (1 mg L^−1^ 2–4D, 0.5 mg L^−1^ 6-BA) and incubated in a dark chamber at 24 °C for 36 h. Following this, the incubated calli were transferred to the MS medium for transgenic selection. RT-PCR and qRT-PCR were used to detect transgenic expression in the obtained transgenic lineages.

### 2.8. Statistical Analysis

To identify statistically significant differences between samples, we employed the Student’s *t*-test (* *p* < 0.05, ** *p* < 0.01). In order to determine noteworthy variations across various sets of data, we employed Tukey’s single-factor analysis of variance (ANOVA) using SPSS 20 (IBM Corporation, Chicago, IL, USA). Graphs were generated using GraphPad Prism 9.0 (GraphPad Software, San Diego, CA, USA). All reported values represent the mean ± standard error (SE) derived from three biological replicates.

## 3. Results

### 3.1. Changes in Quality Parameters of Apple Fruit Treated with Exogenous MeJA

The weight loss rate of fruit has a significant impact on the postharvest quality of apples and is a vital indicator of the fruit storage process. The quality loss in fruit soaked in MeJA and control solution increased with a longer storage time (Figure 1B). Compared with the control group’s 4.53% weight loss rate, there was no significant change in the weight loss rate of fruit treated with MeJA, which was 4.46%. As can be seen from Figure 1C, the peak respiration rates appeared simultaneously on day 9 in control and MeJA-treated fruit. The peak respiratory rate of the control group was 27.34 μL kg^−1^ h^−1^, which was 1.26 times that of the MeJA-treated group. MeJA significantly reduced the respiration rate of apples on days 9–18. The respiratory transition was accompanied by a large release of ethylene, so we tested the ethylene release between the two groups. The results showed that the MeJA treatment group also reached the maximum ethylene release of 160.69 μL kg^−1^ h^−1^ on the 9th day. The ethylene release amount in the control group was 196.23 μL kg^−1^ h^−1^ and then decreased (Figure 1D). The MeJA treatment reduced the ethylene release from the apples and effectively improved the quality of the apples. Firmness is one of the important indicators reflecting fruit maturity. During the 18-day storage period, the fruit firmness of both the control and treatment groups decreased steadily (Figure 1E). On the 6th day and 12th to 18th day after treatment, the firmness of the control group was significantly lower than that of the treatment group. Starting on the 6th day, the SSC of fruit treated with MeJA showed a significant increase compared to the control group (Figure 1F). The TA content showed a decreasing trend throughout the storage process (Figure 1G). Compared with the day of sampling (0th day), the TA content of MeJA-treated fruit decreased by 0.14% on the 18th day, while the TA content of the control fruit only decreased by 0.11% on the 18th day. Significant differences were observed in the *L** value of fruit treated with MeJA on days 9–18 after harvest, reaching a maximum of 75.72 (Figure 1H). The increasing trend in the *a** value of untreated fruit was more obvious. The *b** value of both groups showed an increasing trend within 18 days of storage (Figure 1I,J).

### 3.2. Effects of Exogenous MeJA Treatment on Apple Fruit Cell Wall Components

The decrease in apple fruit tissue hardness is often associated with changes in cell wall polysaccharides. This study examined the effects of exogenous MeJA treatment on various apple fruit cell wall components, including WSP, CSP, ISP, HC, cellulose, and CWM. The WSP content of apple fruit generally increased during storage, but this increase was obviously hindered after MeJA treatment (Figure 2A). Starting on the 9th day, the level of WSP content notably rose in the control group, ultimately peaking at 1.84 mg g^−1^ on the 18th day. The CSP content of control group was 1.38 times that of the MeJA group on the 12th day (*p* < 0.05) (Figure 2B). The ISP content showed a steady increase during storage, with MeJA-treated fruit having lower levels compared to the control group (Figure 2C). Hemicellulose (HC) and cellulose contents decreased over time, with the control group showing a more pronounced decrease (Figure 2D–F). Overall, the MeJA treatment appeared to positively influence the fruit quality by affecting cell wall metabolism.

### 3.3. Effects of Exogenous MeJA Treatment on the Activities of Cell Wall-Degrading Enzymes in Apple Fruit

During storage, the PG activity of apple fruit showed a steadily increasing trend (Figure 3A). Starting from the 6th day after MeJA treatment, the PG activity was significantly lower than that of the control group, indicating that the MeJA treatment can effectively increase the PG activity. As can be seen from Figure 3B, on the 18th day, the PME activity of the control group was 387 U kg^−1^, which was significantly higher than the 367.5 U kg^−1^ of the MeJA group (*p* < 0.05). The Cx activity of the MeJA group had a certain upward trend, but it was significantly lower than the control, reaching 1.57 U kg^−1^ on the 18th day (Figure 3C). In addition, the β-gal activity was significantly higher in the MeJA group (Figure 3D).

### 3.4. Effects of Exogenous MeJA Treatment on the Cell Wall- and Ethylene-Related Enzyme Genes

Utilizing our prior transcriptomic data and in conjunction with previous studies, we examined genes related to cell wall hydrolase and ethylene. The relative expressions of nine candidate genes during storage were detected using qRT-PCR (Figure 4). During storage, all nine genes showed a steadily increasing expression pattern. ACC synthases (ACSs) and ACC oxidases (ACOs) are key enzymes for ethylene synthesis, so it is of great significance to detect the relative expressions of *MdACSs* and *MdACOs*. After MeJA treatment, the upward trends of *MdACS1* and *MdACS2* were more gradual than that of the control (*p* < 0.05 or *p* < 0.01). From the 9th to the 18th day, the relative expression levels of MdACO1 and MdACO2 were significantly lower in the MeJA group (*p* < 0.01). From the results, MeJA impacted the expression of *MdACS* and *MdACO* genes. Pectin lyase breaks down pectin, loosening the fruit tissue and promoting the softening of the fruit. As storage progressed, the expression of *MdPL18* in the control group increased sharply. On the 18th day of storage, the expression level of the control group was even 3.02 times that of the treatment group. The expression of *Mdgal1* in the MeJA group was significantly lower than that in the control group on the 6th day after treatment (*p* < 0.01), and the relative expression of *Mdgal2* was significantly lower than that in the untreated group on the 12th day of storage (*p* < 0.01). The relative expression of *MdPG1* began to show significant differences on the 9th day (*p* < 0.05). These results indicate that MeJA has an effect on the expression of cell wall- and ethylene-related enzyme genes.

### 3.5. Correlation between the Candidate Gene MdPL18 (MD16G1070600) and Fruit Quality

During storage, we observed that the expression level of *MdPL18* in the control group was significantly higher than that in the MeJA group, and *MdPL18* showed a high correlation with various cell wall components (Appendix A). Therefore, we took *MdPL18* as a candidate gene in this study. Firstly, the CDS of *MdPL18* was cloned and isolated from the pericarp of a ‘Ruixue’ apple in this study, and then the amino acid sequence encoded by *MdPL18* was analyzed with the corresponding sequences of other species using the method of constructing a phylogenetic gene tree. The results showed that the amino acid sequence encoded by *MdPL18* had the highest homology with the PLs of Pyrus x bretschneideri, Malus sylvestris, and Prunus dulcis (Appendix A). In addition, their protein structural domains were analyzed in this study using MEME [31] (Appendix A). There were significant differences in the gene expression levels of the *MdPL18* gene in different tissues (root, stem, leaf, flower, peel, and flesh) of the ‘Ruixue’ apple (Appendix A). Firstly, the highest expression level of the *MdPL18* gene was found in the flesh, up to 34-fold, followed by the peel, stem, and leaf, while it was hardly expressed in the flower and root, compared with apple root.

### 3.6. Transient Overexpression of MdPL18 Promotes Ripening and Softening of ‘Ruixue’ Fruit

Transient overexpression experiments were conducted using ‘Ruixue’ fruit due to the difficulty of producing stably transformed perennial apple fruit. The fruit was harvested 2 weeks before commercial ripening (Figure 5A). The results showed that the *MdPL18* gene expression level was significantly increased, nearly 2-fold, in fruit injected with *MdPL18*-pC2300 compared to the null control, while the *MdPL18* gene expression level was significantly reduced in fruit injected with *MdPL18*-pTRV2 (Figure 5B). In addition, we measured cell wall-related substances in ‘Ruixue’ fruit that transiently overexpressed and transiently silenced the *MdPL18* gene. Compared with the control, the WSP content of ‘Ruixue’ fruit with transient silencing of *MdPL18* was significantly lower than that of the control (*p* < 0.01) (Figure 5C), and the CSP and ISP contents of the fruit showed the same expression trend (*p* < 0.01), with the CSP and ISP contents being only 7.26 and 3.59 mg g^−1^ (Figure 5D,E). In contrast, in the *MdPL18*-pC2300 group, the fruit WSP, CSP and ISP contents were 1.44-fold, 1.53-fold, and 1.48-fold higher than those in the control group, respectively. In addition, in the determination of the HC content, cellulose content, and CWM content of transiently transformed fruit, it was found that transient silencing of the *MdPL18* gene increased the content of the above-three cell wall-related substances, and the transient overexpression of the *MdPL18* gene decreased the content of the substances (*p* < 0.01) (Figure 5F–H).

### 3.7. Effect of Stable Overexpression of MdPL18 in Apple Calli on Cell Wall Material Content

To further determine the effect of *MdPL18* on apples, apple callus lines stably overexpressing *MdPL18* were formed in this study (Figure 6A). Firstly, the transgenic ‘Orin’ calli were characterized (Figure 6B), and the gene expression level of *MdPL18* in apple calli overexpressing *MdPL18* was found to be high in OE2, OE4, and OE7 compared to wild-type ‘Orin’ calli; these were significantly increased by 11.92-fold, 14.93-fold, and 14.99-fold, respectively (Figure 6C). Fruit texture changes are the manifestation of changes in its cell wall structure and components. The degradation of pectin, as a major component of the cell wall, is the key to fruit softening, and the hydrolysis of galactose on the side chain of pectin by β-galactosidase (β-gal) is one of the most important causes of fruit softening. Therefore, we examined the β-gal of apple calli, and we found that the β-gal in the overexpressed calli was significantly higher than in the WT, up to 18.09 μmol min^−1^ g^−1^ (Figure 6D). Overall, these results suggest that overexpression of *MdPL18* promotes apple callus ripening.

## 4. Discussion

MeJA treatment can increase the endogenous ethylene production in the fruit and promote the coloring and ripening of the fruit [26]. In our investigation, we observed that the MeJA application expedited the rate of respiration in the ‘Ruixue’ fruit. Research on strawberries has shown that treating them with MeJA can inhibit the growth and maturation of the fruit [18]. The application of MeJA resulted in the elevation of gene expression in those related to pigment metabolism, sugar metabolism, fruit softening, and hormone metabolism. Additionally, MeJA application increased the levels of JA, anthocyanin, and the sugar content [32]. In plums, MeJA significantly increased ethylene production and the respiration rate. MeJA treatment significantly reduced the *L** and hue angle of the fruit [33]. MeJA significantly increased the total phenolic content but decreased the titratable acidity [34]. Studies on tomatoes showed that MeJA accelerated the production of endogenous ethylene in the fruit and resulted in early tomato fruit coloring periods [35]. In apples, MeJA treatment at low concentrations increased ethylene production in the fruit, while higher concentrations decreased ethylene production. Kondo found that MeJA treatment of ‘Golden Crown’ apples accelerated the production of ethylene [36]. During plum storage, MeJA treatment resulted in significant differences in the *L**, color and hue angle values of fruit from the control group [34]. In this study, the *L** values of fruit treated with MeJA were significantly higher than those of the untreated group on days 9–18 after harvest, up to 75.72. Our research shows that exogenous MeJA treatment could alleviate the softening of the apple fruit and improve the fruit quality. This result is consistent with studies on strawberry and peach fruits [17,37]. SSC and TA have a critical effect on the fruit flavor during apple fruit development and are important indicators used as a means of evaluating fruit storage quality. In this study, we found that postharvest MeJA treatment significantly increased the SSC content of ‘Ruixue’ apples and maintained a better fruit quality. MeJA treatment was found to significantly increase the SSC content of fruits in apricot and blueberry studies [38]. With an overall decreasing trend throughout the postharvest apple ripening stage, MeJA treatment significantly contributed to the decrease in TA content, which is the same as the findings in pear fruits [39].

The results showed that there was a trend towards a significant decrease in the WSP content, CSP content, and ISP content of the fruit after exogenous MeJA treatment for 6 d. PG activity is an important factor in the breakdown and conversion of pectin. It is responsible for the depolymerization and solubilization of pectin. However, for PG to effectively break down pectin, the pectin must first undergo demethylation by pectin methyl esterase. This process is essential for the efficient depolymerization of pectin. The inhibition of β-gal activity during the early stage of ripening has been found to have a significant impact on fruit softening. This indicates that the removal of pectin galactose side chains plays a crucial role in the cell wall changes that result in the loss of firmness during ripening [40]. In this study, we found that PG, β-Gal, and other related enzyme activities in exogenous MeJA-treated apple fruit were consistently lower than in the control throughout the postharvest apple ripening stage, indicating that MeJA-treated apple fruit restrained the production rate of cell wall-related enzyme activities throughout the storage period.

The silencing of *MdACS1* in the apple resulted in a decrease in ethylene production in the fruit [41]. Previous studies have shown that *MdACS* is responsible for system Ⅱ ethylene biosynthesis during fruit ripening [41]. In this study, the expressions of *MdACS1* and *MdACS2* were significantly up-regulated in ‘Ruixue’ fruit during postharvest storage, and the expression of the treated group was significantly lower than that of the control group starting on the 12th d after MeJA treatment (*p* < 0.01). According to reports, system II might utilize *MdACO1* and *MdACO2* as pivotal elements in the synthesis of ethylene [42]. In the present study, it was found that the expressions of *MdACO1* and *MdACO2* were significantly down-regulated following MeJA treatment compared with control. The expression levels of MdPL1 and *MdPL18* increased over time during storage. Following MeJA treatment, *MdPL18* expression remained relatively stable in contrast to the sharp increase observed in the control group. This suggests a significant positive correlation between *MdPL18* and pectin degradation. Furthermore, the relative expression levels of the two β-gal candidate genes were also influenced by MeJA, showing a gradual increase.

An association between the expression of pectin lyases and fruit ripening softening has been reported in many fruits, including bananas, tomatoes, and mangoes. However, there is limited research on the functional role of pectin lyase in apple fruit softening. This study focused on *MdPL18*, which showed significant differences in its expression between the control group and the MeJA group and that it was highly correlated with cell wall components. *MdPL18* was selected as a candidate gene for further investigation. The transient overexpression and silencing of the *MdPL18* gene in ‘Ruixue’ apple fruit revealed that overexpression accelerated the degradation of apple fruit cell wall materials and increased the enzyme activities, indicating its involvement in apple fruit softening. Moreover, the overexpression of *MdPL18* in apple fruit calli notably boosted β-Gal activity, further supporting its role in cell wall metabolism and fruit softening.

## 5. Conclusions

Our study demonstrates that treating ‘Ruixue’ apple fruit with appropriate concentrations of exogenous MeJA can effectively reduce fruit softening, mitigate the rapid increase in the respiration rate and ethylene release, and effectively preserve the fruit’s quality. The analysis of cell wall components and their associated degradative enzymes further supports the significant ameliorating impact of MeJA on postharvest apple fruit ripening. Moreover, the discovery of the *MdPL18* gene in ‘Ruixue’ apples is noteworthy. The transient overexpression and silencing of *MdPL18* in ‘Ruixue’ apple fruit indicate that this gene contributes to apple fruit softening by facilitating the breakdown of cell wall materials. These findings offer a solid theoretical foundation for improving the postharvest storage practices for apples.

## Figures and Tables

**Figure 1 foods-13-01594-f001:**
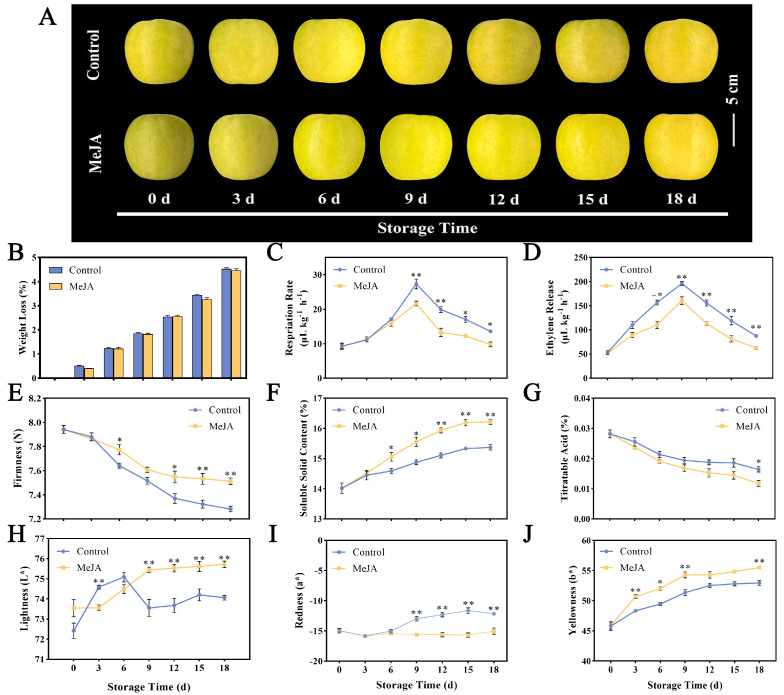
Effects of MeJA on quality traits of ‘Ruixue’ apple fruit. (**A**) Apple fruit morphology, (**B**) weight loss, (**C**) respiratory rate, (**D**) ethylene release, (**E**) firmness, (**F**) SSC, (**G**) TA, (**H**) lightness, (**I**) redness, and (**J**) yellowness (*, *p* < 0.05; **, *p* < 0.01). Error bars show ±SE from three biological replicates.

**Figure 2 foods-13-01594-f002:**
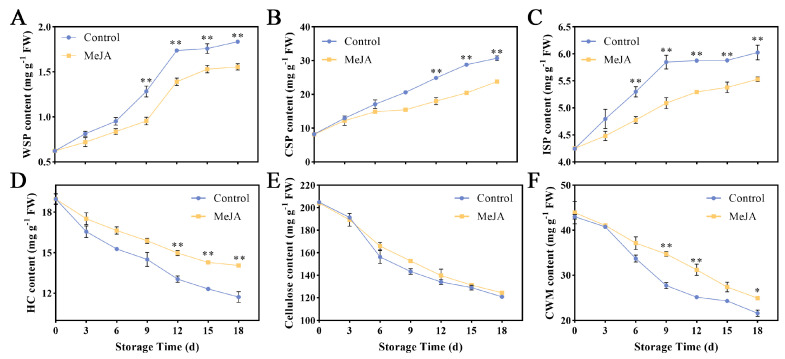
Effect of MeJA on cell wall polysaccharides of ‘Ruixue’ apple fruit. Content of (**A**) WSP, (**B**) CSP, (**C**) ISP, (**D**) HC, (**E**) cellulose, and (**F**) CWM. An asterisk indicates significant differences between control and MeJA fruits (*, *p* < 0.05; **, *p* < 0.01). Error bars show ±SE from three biological replicates.

**Figure 3 foods-13-01594-f003:**
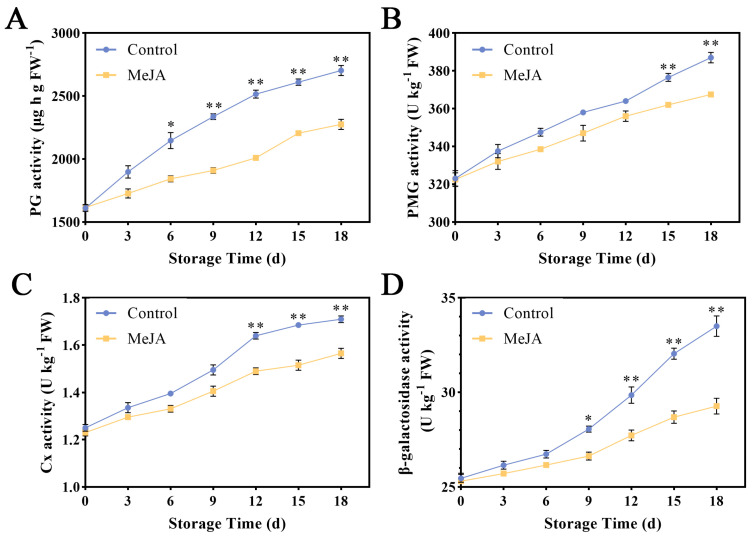
Effect of MeJA on cell wall-degrading enzyme activities in ‘Ruixue’ apple fruit. (**A**) PG, (**B**) PMG, (**C**) Cx, and (**D**) β-gal. An asterisk indicates significant differences between control and MeJA fruits (*, *p* < 0.05; **, *p* < 0.01). Error bars show ±SE from three biological replicates.

**Figure 4 foods-13-01594-f004:**
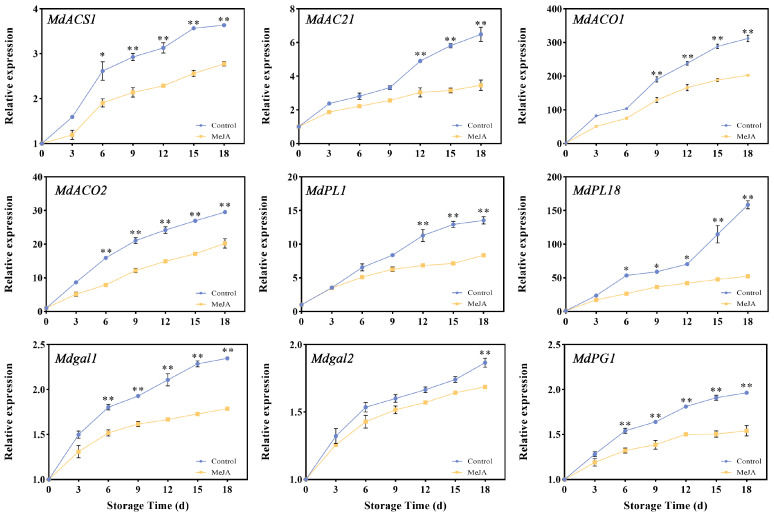
Changes in relative expressions of *MdACS1*, *MdACS2*, *MdACO1*, *MdACO2*, *MdPL1*, *MdPL18*, *Mdgal1*, *Mdgal2*, and *MdPG1* in the exocarp of apples after MeJA treatment. (*, *p* < 0.05; **, *p* < 0.01). Error bars show ±SE from three biological replicates.

**Figure 5 foods-13-01594-f005:**
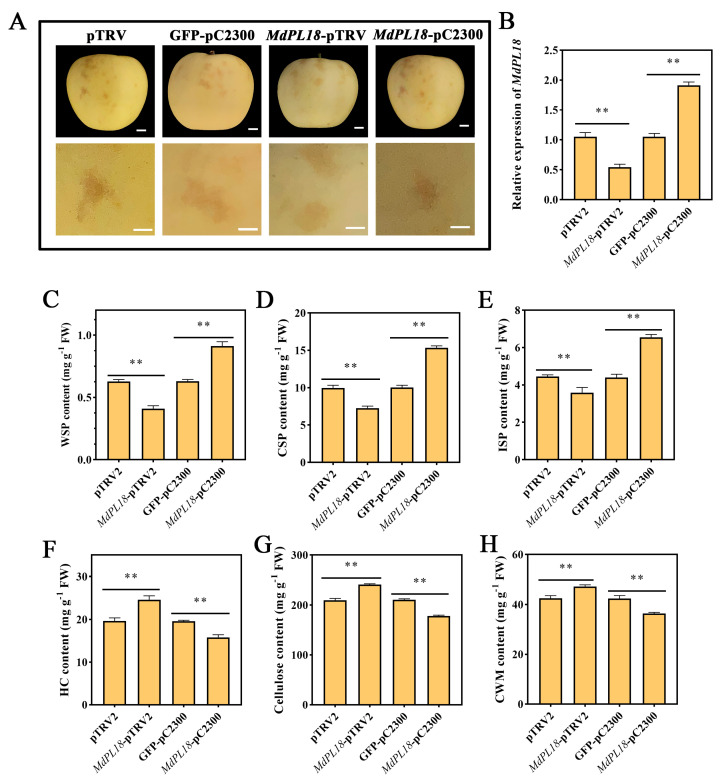
Transient expression analysis and phenotype of *MdPL18* in apple fruit. (**A**) Phenotypes of ‘Ruixue’ apples after transient overexpression and silencing of *MdPL18*, (**B**) gene expression levels of *MdPL18*, and the effects of transient overexpression and silencing of *MdPL18* on (**C**) WSP, (**D**) CSP, (**E**) ISP, (**F**) HC, (**G**) fibrillar, and (**H**) CWM. An asterisk indicates significant differences between control and MeJA fruits (**, *p* < 0.01). Error bars show ±SE from three biological replicates.

**Figure 6 foods-13-01594-f006:**
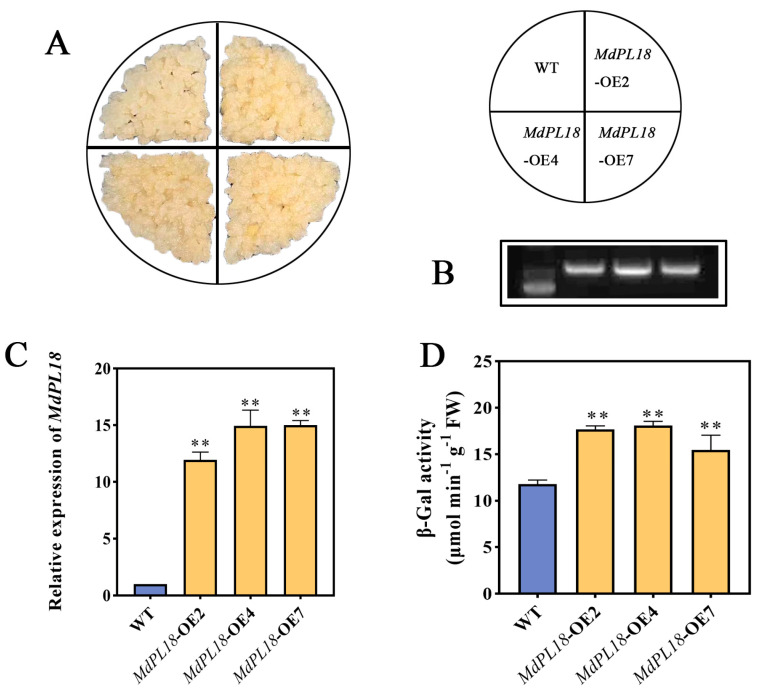
Overexpression of *MdPL18* in apple calli (*Malus domestica* cv. ‘Orin’). (**A**) Phenotypes of apple callus in WT (wild type) and OE- *MdPL18*, (**B**) RT-PCR detection of *MdPL18* expression levels in WT and overexpressing apple calli, (**C**) qRT-PCR to detect the expression level of *MdPL18* in WT and overexpressed apple healing tissues, (**D**) β-galactosidase activity in WT and overexpressed apple calli. An asterisk indicates significant differences between control and MeJA calli (**, *p* < 0.01). Error bars show ±SE from three biological replicates.

## Data Availability

The original contributions presented in the study are included in the article, further inquiries can be directed to the corresponding author.

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
