# Peer review of "Exogenous Methyl Jasmonate (MeJA) Improves ‘Ruixue’ Apple Fruit Quality by Regulating Cell Wall Metabolism"

_foods, 2024, doi:10.3390/foods13111594_

Round 1
Reviewer 1 Report
Comments and Suggestions for Authors
The submitted study is about th ese of MeJA as a helpful postharvest treatment and is focusing in the effect of MeJA on fruit quality and key enzymes and genes related to cell wall metabolism. the general approach to the main objective is sufficient and remarkable finding has been found.
Only some points need to be revised:
-line 57 "defence"
-the word inhibits is used all along the text, check is the correct work because in some cases is not an inhibition effect o the analysis conducted are no related to that.
-Materials and methods, section 2.5; in this section only mentioned the actin as a control gene, however the results in the figure 4 are showed as relative expression, so in this section as to indicated the method used for this (2^ delta?).
-Line (191): Figure 1C mentioned a peak on 12 day, however in the figure this peak appears on the 9 day, actually in the next lines (line 196) say: "also reached".
-Figure 1: only indicates the different line color for control and MeJA, it is necessary to indicate this in all the graphs that appear in this composite figure. This last comment applies to all figures.
Line 235: The MeJa group was 1.38 times that of the MeJa group
-Figure 4. letters from each graphic, and also the indication between control and MeJA.
- the term "enhanced" must be used carefully because in post-harvest we talk about maintaining the quality of the fruit, not really an improvement.
-conclusions
The conclusion is adequate to the results, as a recommendation you could eliminate in line 445 (similar observations were made in transfenic apples and calli) thus directly indicate "these findings offer a solid theoretical foundation for improving..... for apples"
Author Response
Dear Editors and Reviewers:
Thank you for your letter and for the reviewers’ comments concerning our manuscript entitled “Exogenous methyl jasmonate (MeJA) improves ‘Ruixue’apple fruit quality by regulating cell wall metabolism” (ID: foods-2977015). Those comments are all valuable and very helpful for revising and improving our paper, as well as the important guiding significance to our researches. We have studied comments carefully and have made correction which we hope meet with approval. Revised portion are marked in red in the paper. The main corrections in the paper and the responds to the reviewer’s comments are as flowing: Responds to the reviewer’s comments:
Reviewer #1:
1):We are very sorry for our incorrect writing. We changed it in Line 57.
2):Thank you very much. We changed it in Line 13, Line 248, Line 269.
3):We have made correction according to the Reviewer’s comments. We have re-written this part according to the Reviewer’s suggestion. All gene expression levels were normalized using Mdβ-Actin as the internal reference gene. The relative gene expression was calculated by 2−ΔΔCt. (Line 166-168)
4):Thank you very much. We are very sorry for our incorrect writing. We changed it in Line 199-201.
5):Thanks for your suggestions. We changed it in Line 221 (Figure 1), Line 239 (Figure 2), Line 254 (Figure 3), Line 279 (Figure 4).
6):We are very sorry for our incorrect writing. We changed it in Line 233-234.
7):Thanks for your suggestions. We changed it in Line 279.
8):Thanks for your suggestions. We changed it in Line 371-373.
9):Thank you very much. We changed it in Line 425-426.
Special thanks to you for your good comments.

Reviewer 2 Report
Comments and Suggestions for Authors
The article methyl jasmonate (MeJA) improves 'Ruixue'apple fruit quality by Regulating Cell Wall Metabolism, is interesting since it provides a series of elements to take into account for the harvest and storage of apples, it is important to know if the genetic expression Found in the Ruixue variety it also manifests itself in other varieties grown in other countries. MeJa plays a crucial role in fruit metabolic processes, particularly in fruit softening, respiration rate, and ethylene release, which improves fruit quality and therefore shelf life. The authors also show how MeJA is associated with enzymes that degrade the cell wall; genetics are important in the fruit ripening process. The authors are suggested to improve the conclusions, they have a lot of experimental work to rewrite the conclusions.

The article methyl jasmonate (MeJA) improves 'Ruixue'apple fruit quality by Regulating Cell Wall Metabolism, is interesting since it provides a series of elements to take into account for the harvest and storage of apples, it is important to know if the genetic expression Found in the Ruixue variety it also manifests itself in other varieties grown in other countries. MeJa plays a crucial role in fruit metabolic processes, particularly in fruit softening, respiration rate, and ethylene release, which improves fruit quality and therefore shelf life. The authors also show how MeJA is associated with enzymes that degrade the cell wall; genetics are important in the fruit ripening process. The authors are suggested to improve the conclusions, they have a lot of experimental work to rewrite the conclusions.
Author Response
Dear Editors and Reviewers:
Thank you for your letter and for the reviewers’ comments concerning our manuscript entitled “Exogenous methyl jasmonate (MeJA) improves ‘Ruixue’apple fruit quality by regulating cell wall metabolism” (ID: foods-2977015). Those comments are all valuable and very helpful for revising and improving our paper, as well as the important guiding significance to our researches. We have studied comments carefully and have made correction which we hope meet with approval. Revised portion are marked in red in the paper. The main corrections in the paper and the responds to the reviewer’s comments are as flowing: Responds to the reviewer’s comments:
Reviewer #2:
1): Thank you very much. We are very sorry for our incorrect writing. Line 57, the incorrect writing of “signalling” were corrected as “signaling”.
2): Thank you very much. We are very sorry for our incorrect writing. Line 57, the incorrect writing of “defence” were corrected as “defense”.
3): Thank you very much. Line 70, We've changed “strawberries” to plural.
4): Thank you very much. Line 76, “the” was added.
5): Thanks for your suggestions. We changed it in Line 78.
6): Thank you very much. Line 91, “the” was added.
7): Thank you very much. We are very sorry for our incorrect writing. We changed it in Line 95.
8): Thanks for your suggestions. Line 111-112, “a duration of” was deleted. Line 183, “the purpose of” was deleted.
9): Thank you very much. We changed it in Line 118.
10): Thanks for your suggestions. Line 124, 162, 196, 259, 294, 296, 298, 307, 351, 390 “The” was added.
11): Thank you very much. Line 154, the incorrect writing of “colourimetric method” were corrected as “colorimetric method”
12): Thanks for your suggestions. Line 278, We've changed “cell walls” to plural. Line 356, plums. Line 361, apples. Line 407, bananas, tomatoes and mangoes.
13): Thanks for your suggestions. Line 285, “a” was added.
14): Thank you very much. We are very sorry for our incorrect writing. Line 289 and 293, the incorrect writing of “analysed” were corrected as “analized”.
15): Thank you very much. We are very sorry for our incorrect writing. Line 350, 360 and 365, the incorrect writing of “colour” were corrected as “color”.
16): Thank you very much. We are very sorry for our incorrect writing. Line 370, the incorrect writing of “flavour” were corrected as “flavor”.
17): Thanks for your suggestions. Line 381, the incorrect writing of “depolymerisation and solubilisation” were corrected as “depolymerization solubilization”
Special thanks to you for your good comments.

Reviewer 3 Report
Comments and Suggestions for Authors
The manuscript about influence of MeJA on apple fruit quality is interesting and I think that the result of research can be implemented in the food industry.
However, the paper requires some corrections, mainly related to the edition and the publisher's requirements:
- Please consider adding a list of abbreviations used in the manuscript. Some of abbreviations in the text when used for the first time are not explained (eg. Line 88).
- Line 23: maybe “belongs to” instead “is a member” would be more correct
- Please improve the introduction. Consider changing the position of the pectin lyase paragraph to appear after the ethylene paragraph.
- Line 66: please consider changing “restrain” into “inhibit”
- The aim of the work is mention in two paragraphs. First time in paragraph starting in Line 85, and second in Line 96.
- Sentence in Line 93-95 sounds like conclusion.
- Please make changes according to publishers guidelines, especially in the case of references, eg. Line 110 should be “Lv et al. [26]”, list of references has double numeration
- Paragraph “experimental treatments and plan material should be orderly: first please give info about preparation of solutions then about its application. Triton X-100 is a trade mark, so please rewrite it properly
- Please provide more specific info about apparatus in methods, and conditions of measurements
- Line 132: “the hardness of apple pulp” suggests that authors checked this parameter in homogenised/crushed/gratted apples, if so please describe in details the pulp preparation and the hardness measurement, if not please rewrite so the reader will have no doubts about the examined material.
- Line 136: rewrite “yang’s procedure”, eg. “as described by Yand et al. [27]”, same in line 139
- How did authors prepare the apple pulp? Please add proper paragraph with details of procedure and apparatus.
- Please check carefully the manuscript in the case of italic, superscript and subscript, units, double breaks (eg. Line 164, 166 and so on).
- Please explain why you used different apple in the case of overexpression vector (paragraph 2.7). This should be mention in the method.
- In the result authors mention about the moisture content, but there is no procedure describe in paragraph “materials and methods”. Please add proper paragraph.
- Line 198-199: Authors say that MeJA treatment can extend the storage period, but experimental storage period was only 18 day long, so this conclusion is far-fetched.
- The figures 1, 2, and 4 are difficult to read, please increase their resolution, also please add axis descriptions and a legend.
- Line 305: please changed www into reference
- Authors refer to supplementary material, but the is not any.
- Authors Contributions should be describe with abbreviations
- Please add “Data Availability Statement”
- Please rewrite references according to publishers guidelines: position 1 and 39.
Author Response
Dear Editors and Reviewers:
Thank you for your letter and for the reviewers’ comments concerning our manuscript entitled “Exogenous methyl jasmonate (MeJA) improves ‘Ruixue’apple fruit quality by regulating cell wall metabolism” (ID: foods-2977015). Those comments are all valuable and very helpful for revising and improving our paper, as well as the important guiding significance to our researches. We have studied comments carefully and have made correction which we hope meet with approval. Revised portion are marked in red in the paper. The main corrections in the paper and the responds to the reviewer’s comments are as flowing: Responds to the reviewer’s comments:
Reviewer #3:
1): Thanks for your suggestions. We submitted a list of abbreviations.
Abbreviations:
|
Abbreviation |
Appellation |
|
MeJA |
methyl jasmonate |
|
WSP |
water-soluble pectin |
|
CSP |
chelator-soluble pectin |
|
ISP |
ion-soluble pectin |
|
PME |
pectin methylesterase |
|
PG |
polygalacturonase |
|
β-Gal |
glycosidase |
|
PAL |
phenylalanine ammonia-lyase |
|
POD |
peroxidase |
|
PPO |
polyphenol oxidase |
|
ETH |
Ethylene |
|
SSC |
Soluble solid content |
|
TA |
titrable acidity |
|
HC |
hemicellulose |
|
CWM |
cell wall material |
|
PMG |
Polymethylgalacturonase |
|
Cx |
Cellulase |
We changed it in Line 457.
2): Thank you very much. We are very sorry for our incorrect writing. Line 23, the incorrect writing of “is a member” were corrected as “belongs to”.
3): Thanks for your suggestions. We changed it in Line 33-40, Line 41-53.
4): Thank you very much. We changed it in Line 66.
5): We have made correction according to the Reviewer’s comments. We have re-written this part according to the Reviewer’s suggestion As Reviewer suggested that. Line 86-87. “In this research, the effects of exogenous MeJA on fruit quality indicators such as firmness, SSC, TA, and ETH production of 'Ruixue' apple fruit were studied.”
6): Thanks for your suggestions. “In addition, MdPL18 not only directly affects the transcription of fruit ripening-related genes, but also affects cell wall metabolism-related substances, thereby further affecting fruit ripening.” was deleted.
7): Thanks for your suggestions. We changed it in Line 106 and 134.
8): Thank you very much. We have made correction according to the Reviewer’s comments. We have re-written this part according to the Reviewer’s suggestion As Reviewer suggested that “The preparation of the MeJA solution involved dissolving 95% MeJA (Solarbio, Beijing, China) in deionized water with a 0.077% (v/v) triton x-100 solution. The first group was immersed in a solution of 10 μM MeJA for 10 minutes. The control group, on the other hand, was immersed in deionized water with a 0.077% (v/v) triton x-100 solution for 10 minutes.”. We changed it in Line 109-113.
9): Thanks for your suggestions. We changed it in Line 124-127, Line 134-144, 146-148.
10): Thank you very much. We are very sorry for our mischaracterization. We changed it in Line 130, 135, 147, 162, 207-208.
11): Thanks for your suggestions.
We changed it in Line 133-134. SSC and TA are analyzed as described by yang et al. [27].
Line 146. The procedure to extract cell wall polysaccharides followed Melton et al. [28].
12): Thank you very much. We are very sorry for our mischaracterization. The test material we used was apple fruit rather than apple pulp. We changed it in Line 130,135, 147, 162, 209.
13): Thank you very much. We are very sorry for our incorrect writing. We changed it in Line 174, 175, 217, 219, 232, 310, 357, 365, 392.
14): Thanks for your suggestions. In the transgenic process of Apple calli, Orin calli is usually used for testing [1-3]. We are also studying other varieties of calli, thank you very much for your suggestions.
15): Thank you very much. In this study, we only measured the water loss rate of apples and did not measure the water content of the fruit. In the original text, it is our expression that is wrong. We changed it in Line 195-196.
16): Thank you very much. Line 207-208, “extended the storage period of postharvest apples” was deleted.
16): Thanks for your suggestions. We reuploaded the figure in high resolution and legend.
Figure 1. Effect of MeJA on quality traits of 'Ruixue' apple fruits. (A) Apple fruit morphology, (B) Weight loss, (C) Respiratory rate, (D) Ethylene Release, (E) Firmness, (F) SSC, (G) TA, (H) Lightness, (I) Redness and (J) Yellowness (*, P < 0.05; **, P < 0.01). Error bars show ± SE from three biological replicates.
Figure 2. Effect of MeJA on cell wall polysaccharides of 'Ruixue' apple fruit. Content of (A) WSP, (B) CSP, (C) ISP, (D) HC, (E)cellulose, and (F) CWM. An asterisk indicates significant differences between control and MeJA fruit (*, P < 0.05; **, P < 0.01). Error bars show ± SE from three biological replicates.
Figure 4. Changes in relative expression of MdACS1, MdACS2, MdACO1, MdACO2, MdPL1, MdPL18, Mdgal1, Mdgal2, and MdPG1 in the exocarp of apple after MeJA treatment.
17): Thank you very much. We changed it in Line 292-293, 531: https://meme-suite.org/meme/tools/meme. Available online: May 1, 2024.
18): We have resubmitted the supplementary materials. Thank you very much again.
Figure. S1 Effects of MeJA treatment at different concentrations on quality traits of 'Ruixue' apple fruits. (A) Firmness, (B) Respiratory rate, (C) TA and (D) SSC.
Figure. S2 Correlation analysis between content of WSP (A), ISP (B), CSP (C), HC (D), Cellulose (E), CWM (F) and expression levels of MdPL18.
Figure. S3 Analysis of MdPL18 family phylogeny and protein domains. (A) Maximum-likelihood phylogenetic tree reconstructed with MEGA 7.0, using 1000 bootstrap replicates. (B)Protein motifs. The motifs were detected using the MEME website.
Figure. S4 The relative expression levels of the MdPL18 gene in different organs. (*, P < 0.05; **, P < 0.01). Error bars show ± SE from three biological replicates.
Table. S1 Primer sequences used for qRT-PCR.
|
Primer name |
Forward primer |
Reverse primer |
|
MdACS1 |
AGCCTCTCTAAGGATCTTGG |
TGGTTCTCGGCTATGTAGTTCTT |
|
MdACS2 |
ACGGGATTATTCAGATGGGTC |
TGAGCACTAAGTGGTTGGGAT |
|
MdACO1 |
GACTTGGACTGGGAAAGCAC |
GGAGGGTAGTTGCTGACCTT |
|
MdACO2 |
ACTCATTCATCAGCCCAGCAC |
AAACCCAAACTTTCTTCTCCC |
|
MdPL1 |
CTTACGGGGGAGGAGGGTG |
TGGGCAATGTGCTGGAAGA |
|
MdPL18 |
CATCCTGTAAACCCTAAACCCG |
CTACCACCAAAAATGGAGACCC |
|
Mdgal1 |
GGGAGTTCTTCTGTTGAATGGG |
TTCCGGCATAAGAACAATCG |
|
Mdgal18 |
CGAAATACCCTGTTAGAGTGAGC |
CGATGAATGACTGCCAAGGAA |
|
MdPG1 |
CGACTTCGCCACCACCGT |
CCGAGCCCAAAGCCATAAA |
19): Thanks for your suggestions. We resubmitted an Authors contribution. We changed it in Line 445-450.
Author Contributions:
D.-X.Y.: conceptualization, software, writing—original draft preparation, writing—review and editing. W.-B.: methodology, resources, writing—review and editing. G.-Y.B.: validation, data curation. Y.-X.Q.: formal analysis, visualization, supervision. C.-X.X.: validation. Z.-Y.W.: investigation. Z.-Z.Y.: conceptualization, writing—review and editing, supervision, project administration, funding acquisition. All authors have read and agreed to the published version of the manuscript.
20): Thanks for your suggestions. We changed it in Line 4553-455.
Data Availability Statement
The original contributions presented in the study are included in the article, further inquiries can be directed to the corresponding author.
21): Thank you very much.
We changed it in Line 459-460, Line 554-555.
Zhao, J., Quan, P., Liu, H., Li, L., Xing, L. Transcriptomic and metabolic analyses provide new insights into the apple fruit quality decline during long-term cold storage. J Agr Food Chem, 2020, 68, 4699-4716. https://doi.org/10.1021/acs.jafc.9b07107.
Gupta, R., Mehta, G., Deswal, D., Sharma, S., Singh, A. Cellulases and their biotechnological applications. Biotechnology for Environmental Management and Resource Recovery, 2013, pp 89-106. https://doi.org/10.1007/978-81-322-0876-1_6.
Special thanks to you for your good comments.
Reference:
- Wang, M., Wu, Y., Zhan, W., Wang, H., Chen, M., Li, T. The apple transcription factor MdZF-HD11 regulates fruit softening by promoting Mdβ-GAL18 J. Exp. Bot, 2023, 75, 819-836. https://doi.org/10.1093/jxb/erad441.
- Liu, X., Li, D., Li, Y., Li, S., Zhao, Z. Brassinosteroids are involved in volatile compounds biosynthesis related to MdBZR1 in ‘Ruixue’ (Malus × domestica) fruit. Postharvest Biol. Technol, 2022, 189, 111931. https://doi.org/10.1016/j.postharvbio.2022.111931.
- Wang, H., Zhang, S., Fu, Q., Wang, Z., Liu, X., Zhao, Z. Transcriptomic and metabolomic analysis reveals a protein module involved in preharvest apple peel browning. Plant Physiol, 2023, 00, 1-21. https://doi.org/10.1093/plphys/kiad064.
